# Continual HyperTransformer:
# A Meta-Learner for Continual Few-Shot Learning

**Max Vladymyrov**                                               *mxv@google.com*
*Google Research*

**Andrey Zhmoginov**                                             *azhmogin@google.com*
*Google Research*

**Mark Sandler**                                                 *sandler@google.com*
*Google Research*

**Reviewed on OpenReview:** *https://openreview.net/forum?id=zdtSqZnkx1*

## Abstract

We focus on the problem of learning without forgetting from multiple tasks arriving sequentially, where each task is defined using a few-shot episode of novel or already seen classes. We approach this problem using the recently published HYPERTRANSFORMER (HT), a Transformer-based hypernetwork that generates specialized task-specific CNN weights directly from the support set. In order to learn from a continual sequence of tasks, we propose to recursively re-use the generated weights as input to the HT for the next task. This way, the generated CNN weights themselves act as a representation of previously learned tasks, and the HT is trained to update these weights so that the new task can be learned without forgetting past tasks. This approach is different from most continual learning algorithms that typically rely on using replay buffers, weight regularization or task-dependent architectural changes. We demonstrate that our proposed CONTINUAL HYPERTRANSFORMER method equipped with a prototypical loss is capable of learning and retaining knowledge about past tasks for a variety of scenarios, including learning from mini-batches, and task-incremental and class-incremental learning scenarios.

## 1 Introduction

Continual few-shot learning involves learning from a continuous stream of tasks described by a small number of examples without forgetting previously learned information. This type of learning closely resembles how humans and other biological systems acquire new information, as we can continually learn novel concepts with a small amount of information and retain that knowledge for an extended period of time. Algorithms for continual few-shot learning can be useful in many real-world applications where there is a need to classify a large number of classes in a dynamic environment with limited observations. Some practical applications can include enabling robots to continually adapt to changing environments based on an incoming stream of sparse demonstrations or allowing for privacy-preserving learning, where the model can be trained sequentially on private data sharing only the weights without ever exposing the data.

To tackle this problem, we propose using HYPERTRANSFORMER (HT; Zhmoginov et al. 2022), a recently published few-shot learning method that utilizes a large hypernetwork (Ha et al., 2016) to meta-learn from episodes sampled from a large set of few-shot learning tasks. The HT is trained to directly generate weights of a much smaller specialized Convolutional Neural Network (CNN) model using only few labeled examples. This works by decoupling the domain knowledge model (represented by a Transformer; Vaswani et al. 2017) from the learner itself (a CNN), generated to solve only a given specific few-shot learning problem.

We present a modification to HT method, called CONTINUAL HYPERTRANSFORMER (CHT), that is aimed at exploring the capability of the HT to *sequentially update the CNN weights* with the information from a new task, while retaining the knowledge about the tasks that were already learned. In other words, given the CNN weights $\theta_{t-1}$ generated after seeing some previous tasks $0, \ldots, t-1$ and a description of the new task $t$, the CHT generates the weights $\theta_t$ that are suited for all the tasks $0, \ldots, t$.

In order for the CHT to be able to absorb a continual stream of tasks, we modified the loss function from a cross-entropy that was used in the HT to a more flexible prototypical loss (Snell et al., 2017), that uses prototypes as a learned representation of every class from all the tasks. As the tasks come along, we maintain and update a set of prototypes in the embedding space. The prototypes are then used to predict the class and task attributes for a given input sample.

We evaluate CHT in three realistic scenarios where a continual few-shot learning model like ours might be used: the mini-batch version, where every task consists of the same classes; the lifelong learning version, where classes for all the tasks are drawn from the same overall distribution; and the heterogeneous task semantic version, where every task has its own unique distribution of classes.

We also test CHT in two different continual learning scenarios: task-incremental learning (predicting class attributes using the task information) and class-incremental learning (predicting class attributes without access to task information; also known as lifelong learning). Moreover, we show empirically that a model trained for class-incremental learning can also perform well in task-incremental learning, similar to a model specifically trained for task-incremental learning.

Our approach has several advantages. First, as a hypernetwork, the CHT is able to generate and update the weights of the CNN on the fly with no training required. This is especially useful for applications That require generating a lot of custom CNN models (such as user-specific models based on their private data). A trained Transformer holds the domain world-knowledge and can generalize from limited few-shot observations.

Second, we did not observe catastrophic forgetting in CHT models when evaluating them on up to 5 tasks using OMNIGLOT and TIEREDIMAGENET datasets. We even see cases of the positive backward transfer for smaller generated CNN model, where the performance on a given task actually improves for subsequently generated weights.

Third, while the CHT is trained to optimize for $T$ tasks, the model can be stopped at any point $t \leq T$ during the inference with weights $\theta_t$ that are suited for all the tasks $0 \leq \tau \leq t$. Moreover, the performance of a given weight $\theta_t$ improves when the CHT is trained on more tasks $T$.

Finally, we designed the CHT model to be independent from a specific step and operate as a recurrent system. It can be used to learn a larger number of tasks it was originally trained for.

## 2 Related work

**Few-shot learning** Many few-shot learning methods can be divided into two categories: metric-based learning and optimization-based learning. First, *metric-based* methods (Vinyals et al., 2016; Snell et al., 2017; Sung et al., 2018; Oreshkin et al., 2018) train a fixed embedding network that works universally for any task. The prediction is based on the distances between the known embeddings of the support set and the embeddings of the query samples. These methods are not specifically tailored for the continual learning problem, since they treat every task independently and have no memory of the past tasks.

Second, *optimization-based* methods (Finn et al., 2017; Nichol & Schulman, 2018; Antoniou et al., 2019; Rusu et al., 2019) propose to learn an initial fixed embedding, which is later adapted to a specific task using a few gradient-based steps. They are not able to learn continually, as simply adapting the embedding for a new task will result in the catastrophic forgetting of previously learned information. In addition, in contrast to these methods, our proposed CHT generates the CNN weights directly on the fly, with no additional gradient adaptations required. This means that for a given task we do not have to use second-order derivatives, which improves stability of our method and reduces the size of the computational graph.

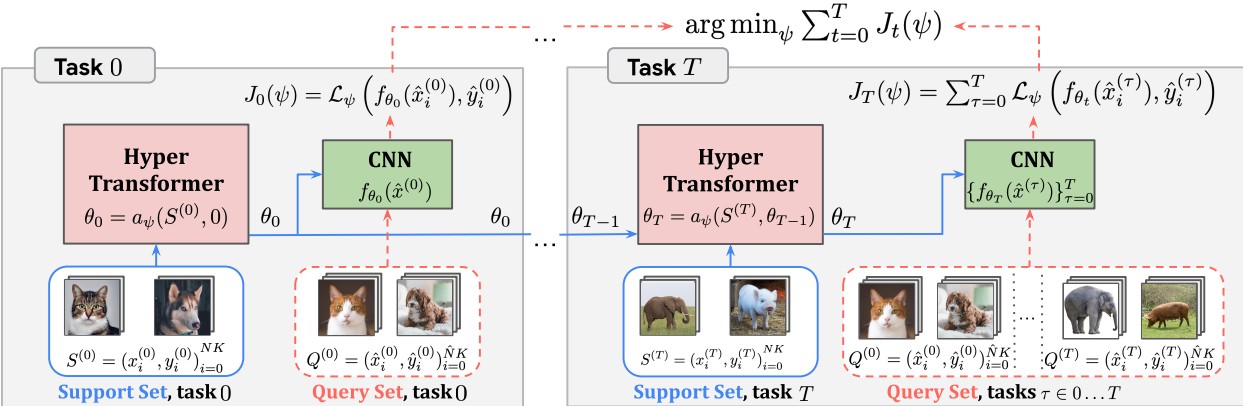

Figure 1: In continual few-shot learning, the model learns from $T$ tasks sequentially. For the first task (task 0), the CNN weights $\theta_0$ are generated using only the support set $S^{(0)}$. For each subsequent task $t$, the CONTINUAL HYPERTRANSFORMER (CHT) uses the support set $S^{(t)}$ and the previously generated weights $\theta_{t-1}$ to generate the weights $\theta_t$. To update the weights $\psi$ of the CHT, the loss is calculated by summing the individual losses computed for each generated weight $\theta_t$ when evaluated on the query set of all the prior tasks $(Q^{(\tau)})_{\tau=0}^T$.

**Continual learning**  Most continual learning methods can be grouped into three categories based on their approach to preventing catastrophic forgetting when learning a new task: rehearsal, regularization and architectural (see Pasunuru et al. 2021; Biesialska et al. 2020 for an overview). *Rehearsal* methods work by injecting some amount of replay data from past tasks while learning the new task (Lopez-Paz & Ranzato, 2017; Riemer et al., 2018; Rolnick et al., 2019; Gupta et al., 2020; Wang et al., 2021a) or distilling a part of a network using task-conditioned embeddings (Mandivarapu et al., 2020; Von Oswald et al., 2019). *Regularization* methods introduce an explicit regularization function when learning new tasks to ensure that old tasks are not forgotten (Kirkpatrick et al., 2017; Zenke et al., 2017). *Architectural* methods modify the network architecture with additional task-specific modules (Rusu et al., 2016), ensembles (Wen et al., 2020) or adapters (Pfeiffer et al., 2020) that allow for separate routing of different tasks.

We believe that our approach requires the least conceptual overhead compared to the techniques above, since it does not impose any additional explicit constraints to prevent forgetting. Instead, we reuse the same principle that made HT work in the first place: decoupling the specialized representation model (a CNN) from the domain-aware Transformer model. The Transformer learns how to best adapt the incoming CNN weights in a way that the new task is learned and the old tasks are not forgotten. In this sense, the closest analogy to our approach would be slow and fast weights (Munkhdalai & Yu, 2017), with the Transformer weights being analogous to the slow weights that accumulate the knowledge and generate CNN weights as fast weights.

**Incremental few-shot learning**  A related, but distinct area of research is incremental few-shot learning (Gidaris & Komodakis, 2018; Ren et al., 2019; Perez-Rua et al., 2020; Chen & Lee, 2020; Tao et al., 2020; Wang et al., 2021b; Shi et al., 2021; Zhang et al., 2021; Mazumder et al., 2021; Lee et al., 2021; Yin et al., 2022). There, the goal is to adapt a few-shot task to an *existing* base classifier trained on a *large* dataset, without forgetting the original data. The typical use case for this would be a single large classification model that needs to be updated once in a while with novel classes. These methods do not work when the existing base classifier is not available or hard to obtain. Another issue is applicability. These methods have to be retrained for each new sequence of tasks, which severely limits their practical application. In contrast, CHT is meta-trained on a large public set of many few-shot classes. After training, it can generate task-dependent $\theta$ in a matter of seconds for many novel sequences.

Consider the following use-case scenario. There is a large public database of many few-shot classes available on the server, and a large number of users with private sequential few-shot data. There is a whole range of applications one can think of in this space, for example smartphone hotword detection or robot calibration

of unique environments in users' homes. Incremental few-shot learning would not be able to either train a reliable base classifier due to the large number of classes on the server, nor effectively deal with user's private data, as it needs to be re-trained each time on the users' devices, which is a lengthy process.

On the other hand, our CONTINUAL HYPERTRANSFORMER can be trained on a large public dataset on the server (with as many classes as needed, as it trains episodically and does not require a single monolithic classifier) and then shipped to devices. Each user can then generate or update their model on the fly without any training required, in a matter of seconds. In the paper we have demonstrated the ability of the CHT to update the user's private classifiers with novel instances of already existing classes (Section 6.1) as well as new classes (Section 6.2 and 6.3) on the fly without retraining the system. This would be impossible or require lengthy retraining with any existing approach.

Perhaps the closest to our setting is the paper by Antoniou et al. (2020) which focuses on the general problem definition of the continual few-shot learning, but falls short of providing a novel method to solve it. Another related method is Wang et al. (2023) which proposes a heterogeneous framework that uses visual as well as additional semantic textual concepts. In our paper we only focus on visual input for model training and predictions.

## 3 Continual few-shot learning

We consider the problem of continual few-shot learning, where we are given a series of $T$ tasks, where each task $t := \{S^{(t)}, Q^{(t)}\}$ is specified via a $K$-way $N$-shot support set $S^{(t)} := (x_i^{(t)}, y_i^{(t)})_{i=0}^{NK}$ and a query set $Q^{(t)} := (\hat{x}_i^{(t)}, \hat{y}_i^{(t)})_{i=0}^{\hat{N}K}$, where $K$ is the number of classes in each task, $N$ is the number of labeled examples for each class, and $\hat{N}$ (typically $\hat{N} \gg N$) is the number of query examples to be classified.

We assume that the classes composing each individual task are drawn from the same distribution uniformly at random without replacement. However, we consider different ways in which classes for different tasks are chosen. First, each task may include exactly the same set of classes. This is similar to mini-batch learning with $T$ iterations, where each batch contains exactly $N$ examples of each of $K$ classes[1]. Second, each task might include a different set of classes, but drawn from the same overall distribution of classes. This corresponds to a lifelong learning scenario, where tasks can be thought of as observations that allow us to learn more about the world as we encounter new classes during the inference. Finally, each task might have its own unique semantic meaning and the classes for different tasks are drawn from different distributions. We will evaluate all of these scenarios in our experiments.

Figure 1 illustrates the process of learning of a continual few-shot problem. For each of the tasks $t \in 0, \ldots, T$, a learner $a_\psi$ (parameterized by $\psi$) needs to produce CNN weights $\theta_t$ based on the support set $S^{(t)}$ of task $t$ and previously generated weights $\theta_{t-1}$ (except for the first task, where $\theta_t$ is generated only using $S^{(0)}$):

$$\theta_t := a_\psi \left( S^{(t)}, \theta_{t-1} \right), \tag{1}$$

such that $\theta_t$ can predict the classes from all the tasks $\tau \in 0, \ldots, t$. Notice that when learning from task $t$, the learner does not have access to the support set of past tasks and must rely solely on the input weights $\theta_{t-1}$ as a source of information from previous tasks.

After the weights $\theta_t$ are generated, we can use the query set $Q^{(\tau)}$ of all tasks $\tau \in 0, \ldots, t$ to evaluate the prediction quality of the $\theta_t$ and calculate the loss $\mathcal{L}_\psi$ with respect to the learner parameters $\psi$. In this work, we consider two types of predictions given the weights $\theta_t$:

- *Task-incremental learning*, in which the goal is to identify the class attribute given the sample and its task attribute: $p(\hat{y} = k | \hat{x}, \tau)$.

- *Class-incremental learning*, in which the goal is to identify both class and task attributes of the samples: $p(\hat{y} = k, \tau | \hat{x})$.

---

[1]This scenario does not require continual learning per se, as the classes do not change between the tasks.

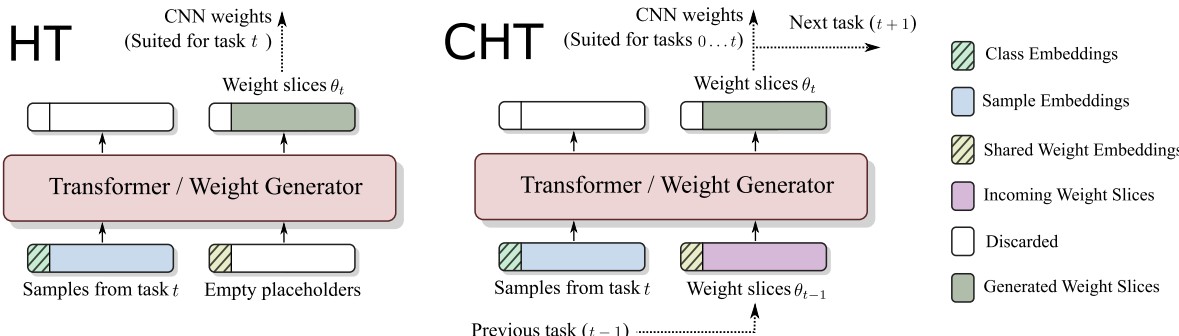

Figure 2: The information flow of the HYPERTRANSFORMER (HT) model (*left*) compared to the proposed CONTINUAL HYPERTRANSFORMER (CHT) model (*right*). In the original HT, the input weight embeddings are initialized with empty placeholders. In contrast, the proposed CHT model incorporates information from past tasks when generating weights for the current task. The weight slice information from previously learned tasks is passed as input to the new iteration of the CHT. The CHT uses the support set for the current task and the input weight information to generate the weights. This allows the CHT to retain knowledge about past tasks and avoid forgetting when learning new tasks.

Finally, we can test the performance of the trained model $a_\psi$ on episodes sampled from a holdout set of classes $\mathcal{C}_{\text{test}}$. Notice that, in general, the total number of tasks for the test $T_{test}$ might be different from $T$.

## 4 Continual HyperTransformer

Notice that for $T = 1$, the continual learning problem above reduces to a standard few-shot learning problem defined by a single few-shot learning task $t_0 = \{S^{(0)}, Q^{(0)}\}$. One method that has been effective in solving this type of problem is HYPERTRANSFORMER (HT, Zhmoginov et al., 2022) that uses a self-attention mechanism to generate CNN weights $\theta$ directly from the support set of the few-shot learning problem (see Figure 2, left).

We first describe the HT architecture. CNN weights are constructed layer by layer using the embeddings of the support set and the activations of the previous layer. After the weights have been generated, the cross-entropy loss $\mathcal{L}_\psi (f_\theta(\hat{x}), \hat{y})$ is calculated by running the query set $(\hat{x}, \hat{y})$ through the generated CNN.

To encode the knowledge of the training task distribution HT is using Transformer model that generates the resulting CNN layer-by-layer. The input of the Transformer consists of concatenation of image embeddings, activation embeddings and support sample labels. The image embedding is produced by a convolutional feature extractor, this time shared among all the layers. The activation embedding is given by another feature extractor that is applied to the activation of the previous label (or the input, in case of the first layer). Together, the parameters of Transformer, image embedding and activation embedding constitute the HT parameters $\psi$.

Our proposed CONTINUAL HYPERTRANSFORMER (CHT) naturally extends HT to handle a continual stream of tasks by using the generated weights from already learned tasks as input weight embeddings into the weight generator for a new task (see Figure 2, right). In this way, the learned weights themselves act as both the input and the output of the CHT, performing a dual function: storing information about the previous tasks *as well as* serving as the weights for the CNN when evaluating on tasks that have already been seen.

For each task $t$, the CHT takes as input the support set of that task $S^{(t)}$ as well as the weights from the previous tasks $\theta_{t-1}$, and generates the weights using the equation (1) that are suited for all the tasks $\tau \in 0, \ldots, t$. Therefore, for each step $t$ we want to minimize the loss on the query sets of every task up to $t$:

$$J_t(\psi) = \sum_{\tau=0}^{t} \mathcal{L}_\psi \left( f_{\theta_t}(\hat{x}^{(\tau)}), \hat{y}^{(\tau)} \right). \tag{2}$$

The overall loss function is simply the sum of the losses for all tasks:

$$\arg\min_{\psi} \sum_{t=0}^{T} J_t(\psi). \tag{3}$$

The CHT generates a sequence of weights $\{\theta_\tau\}_{\tau=0}^{t}$, such that each weight is suited for all tasks up to the current task: $\theta_0$ performs well only on task 0, $\theta_1$ performs well on tasks 0 and 1, and so on. This allows the model to effectively learn and adapt to a stream of tasks, while also maintaining good performance on previously seen tasks.

This design allows for a "preemptive" approach to continual learning, where the CHT model can be trained on $T$ tasks, and run for any number of tasks $\tau < T$, producing well-performing weights $\theta_\tau$ for all the tasks seen up to that point. An alternative approach would be to specify the exact number of tasks in the sequence in advance, and only consider the performance after the final task $T$. This would correspond to minimizing only the last term $J_T(\psi)$ in the equation (3). However, in our experiments, we did not observe any significant improvement using this approach compared to the one we have described above.

Another desirable property of the proposed CHT architecture is its ability to be recurrent. The parameters of the HT do not depend on task information, and only take the weights $\theta$ and the support set as input. This means that it is not only possible to preempt CHT at some earlier task, but also extend the trained model to generate weights for additional tasks beyond the ones it was trained. We will demonstrate this ability in the experimental section.

---

**Algorithm 1** Class-incremental learning using HYPERTRANSFORMER with Prototypical Loss.

**Input:** $T$ randomly sampled $K$-way $N$-shot episodes: $\{S^{(t)}; Q^{(t)}\}_{t=0}^{T}$.
**Output:** The loss value $J$ for the generated set of tasks.

1: $J \leftarrow 0$                                                        ▷ Initialize the loss.
2: $\theta_{-1} \leftarrow 0$                                                      ▷ Initialize the weights.
3: **for** $t \leftarrow 0$ to $T$ **do**
4:     $\theta_t \leftarrow a_\psi(S^{(t)}, \theta_{t-1})$                              ▷ Generate weight for current task.
5:     **for** $k \leftarrow 0$ to $K$ **do**                ▷ Compute prototypes for every class of the current task.
6:         $c_{tk} \leftarrow \frac{1}{N} \sum_{(x,y) \in S^{(t)}} f_{\theta_t}(x) \mathbf{1}_{y=k}$
7:     **end for**
8:     **for** $\tau \leftarrow 0$ to $t$ **do**          ▷ Update the loss with every seen query set using the equation (6).
9:         **for** $k \leftarrow 0$ to $K$ **do**
10:             $J \leftarrow J - \sum_{(\hat{x},\hat{y}) \in Q^{(\tau)}} \log p(\hat{y} = k, \tau|\hat{x}) \mathbf{1}_{\hat{y}=k}$
11:         **end for**
12:     **end for**
13: **end for**

---

## 4.1 Prototypical loss

The last element of the algorithm that we have left to discuss is the exact form of loss function $\mathcal{L}_\psi(\cdot)$ in the equation (2). The original HT used the cross-entropy loss, which is not well suited for continual learning because the number of classes that it predicts is tied to the number of parameters in the head layer of the weights $\theta$. This means that as the number of tasks increases, the architecture of CNN needs to be adjusted, which goes against our design principle of using a recurrent CHT architecture. Another option would be to fix the head layer to the $K$-way classification problem across all the tasks and only predict the class information within tasks (a problem known as domain-incremental learning; Hsu et al., 2018). However, this would cause classes with the same label but different tasks to be minimized to the same location in the embedding space, leading to collisions. Additionally, since class labels are assigned at random for each training episode, the collisions would occur randomly, making it impossible for CHT learn the correct class assignment. In the Appendix A.1, we show that the accuracy of this approach decreases dramatically as the number of tasks increases and becomes impractical even for just two tasks.

To make the method usable, we need to decouple the class predictions of every task while keeping the overall dimensionality of the embedding space fixed. One solution is to come up with a fixed arrangement of $TK$ points, but any kind of such arrangement is suboptimal because it is not possible to place $TK$ points equidistant from each other in a fixed-dimensional space for large $T$. A much more elegant solution is to learn the location of these class prototypes from the support set itself, e.g. with a prototypical loss (Snell et al., 2017). The prototypes are computed by averaging the embeddings of support samples from a given class $k$ *and* task $\tau$:

$$c_{\tau k} := \frac{1}{N} \sum_{(x,y) \in S^{(\tau)}} f_{\theta_\tau}(x) \mathbf{1}_{y=k}. \tag{4}$$

We can use the prototypes in two different continual learning scenarios. First, for the *task-incremental learning*, we are assumed to have access to the task we are solving and need to predict only the class information. The probability of the sample belonging to a class $k$ given the task $\tau$ is then equal to the softmax of the $\ell_2$ distance between the sample and the prototype normalized over the distances to the prototypes from all the classes from $\tau$:

$$p(\hat{y} = k | \hat{x}, \tau) := \frac{\exp(-\|f_{\theta_t}(\hat{x}) - c_{\tau k}\|^2)}{\sum_{k'} \exp(-\|f_{\theta_t}(\hat{x}) - c_{\tau k'}\|^2)}. \tag{5}$$

Second, for more general *class-incremental learning*, we need to predict class attributes across all seen tasks. The probability of a sample belonging to class $k$ of task $\tau$ is equal to the softmax of the $\ell_2$ distance between the sample and the prototype, normalized over the distances to the prototypes from all classes for all tasks:

$$p(\hat{y} = k, \tau | \hat{x}) := \frac{\exp(-\|f_{\theta_t}(\hat{x}) - c_{\tau k}\|^2)}{\sum_{\tau' k'} \exp(-\|f_{\theta_t}(\hat{x}) - c_{\tau' k'}\|^2)}. \tag{6}$$

The final loss function is given by minimizing the negative log probability of the chosen softmax over the query set. The pseudo-code for the entire CHT model is described in Algorithm 1.

Empirically, we noticed that the CHT models trained with the class-incremental learning objective (6) perform equally well in both class-incremental and task-incremental settings, while models trained with the task-incremental objective (5) perform well only in the task-incremental setting and rarely outperform models trained with the equation (6). Therefore, we will focus on CHT models trained with the equation (6) and evaluate them for both task- and class-incremental learning scenarios.

Notice that the prototypes are computed using the current weights $\theta_\tau$ in the equation (4) for task $\tau$, but they are used later to compare the embeddings produced by subsequent weights $\theta_t$ in equation (6). Ideally, once the new weights $\theta_t$ are generated, the prototypes should be recomputed as well. However, in true continual learning, we are not supposed to reuse the support samples after the task has been processed. We have found that freezing the prototypes after they are computed provides a viable solution to this problem, and the difference in performance compared to recomputing the prototypes every step is marginal.

Finally, we want to highlight an important use-case where recomputing the prototypes might still be possible or even desirable. The weights $\theta_t$ are not affected by this issue and are computed in a continual learning manner from the equation (1) without using information from the previous task. The support set is only needed to update the prototypes through generated weights, which is a relatively cheap operation. This means that it is possible to envision a privacy-preserving scenario in which the weights are updated and passed from client to client in a continual learning manner, and the prototypes needed to "unlock" those weights belong to the clients that hold the actual data.

## 5 Connection Between Prototypical Loss and MAML

While the core idea behind the prototypical loss is very natural, this approach can also be viewed as a special case of a simple 1-step MAML-like learning algorithm. This can be demonstrated by considering a simple classification model $\boldsymbol{q}(x; \phi) = s(\boldsymbol{W} f_\theta(x) + \boldsymbol{b})$ with $\phi = (\boldsymbol{W}, \boldsymbol{b}, \theta)$, where $f_\theta(x)$ is the embedding and $s(\cdot)$

is a softmax function. MAML algorithm identifies such initial weights $\phi^0$ that any task $\tau$ with just a few gradient descent steps initialized at $\phi^0$ brings the model towards a task-specific local optimum of $\mathcal{L}_\tau$.

Notice that if any label assignment in the training tasks is equally likely, it is natural for $\boldsymbol{q}(x; \phi^0)$ to not prefer any particular label over the others. Guided by this, let us choose $\boldsymbol{W}^0$ and $\boldsymbol{b}^0$ that are *label-independent*. Substituting $\phi = \phi^0 + \delta\phi$ into $\boldsymbol{q}(x; \phi)$, we then obtain

$$
q_\ell(x; \phi) = q_\ell(x; \phi^0) + s'_\ell(\cdot)\left(\delta\boldsymbol{W}_\ell f_{\theta^0}(x) + \delta b_\ell + \boldsymbol{W}_\ell^0 \frac{\partial f}{\partial \theta}(x; \theta^0)\delta\theta\right) + O(\delta\phi^2),
$$

where $\ell$ is the label index and $\delta\phi = (\delta\boldsymbol{W}, \delta\boldsymbol{b}, \delta\theta)$. The lowest-order label-dependent correction to $q_\ell(x; \phi^0)$ is given simply by $s'_\ell(\cdot)(\delta\boldsymbol{W}_\ell f_{\theta^0}(x) + \delta b_\ell)$. In other words, in the lowest-order, the model only adjusts the final logits layer to adapt the pretrained embedding $f_{\theta^0}(x)$ to a new task.

For a simple softmax cross-entropy loss (between predictions $\boldsymbol{q}(x)$ and the groundtruth labels $y$), a single step of the gradient descent results in the following logits weight and bias updates:

$$
\delta\boldsymbol{W}_{i,\cdot} = \frac{\gamma}{n}\sum_{(x,y)\in S}\left(\mathbf{1}_{y=k} - \frac{1}{|C|}\right)f_{\theta^0}(x), \qquad \delta b_k = \frac{\gamma}{n}\sum_{(x,y)\in S}\left(\mathbf{1}_{y=k} - \frac{1}{|C|}\right), \tag{7}
$$

where the $1/|C|$ term results from normalization in the softmax operation. Here $\gamma$ is the learning rate, $n$ is the total number of support-set samples, $|C|$ is the number of classes and $S$ is the support set. In other words, we see that the label assignment imposed by $\delta\boldsymbol{W}$ and $\delta\boldsymbol{b}$ from the equation (7) effectively relies on computing a dot-product of $f_{\theta^0}(x)$ with "prototypes" $c_k := N^{-1}\sum_{(x,y)\in S}f_{\theta^0}(x)\mathbf{1}_{y=k}$.

## 6 Experiments

Most of our experiments were conducted using two standard benchmark problems using OMNIGLOT and TIEREDIMAGENET datasets. The generated weights for each task $\theta_t$ are composed of four convolutional blocks and a single dense layer. Each of the convolutional blocks consist of a $3\times3$ convolutional layer, batch norm layer, ReLU activation and a $2\times2$ max-pooling layer. Preliminary experiments have showed that batch norm is important to achieve the best accuracy. For OMNIGLOT we used 8 filters for convolutional layers and 20-dim FC layer to demonstrate how the network works on small problems, and for TIEREDIMAGENET we used 64 filters for convolutional and 40-dim for the FC layer[2] to show that the method works for large problems as well. The models were trained in an episodic fashion, where the examples for each training iteration are sampled from a given distribution of classes. The reported accuracy was calculated from 1024 random episodic evaluations from a separate test distribution, with each episode run 16 times with different combinations of input samples.

For the HT architecture, we tried to replicate the setup used in the original paper as closely as possible. We used a 4-layer convolutional network as a feature extractor and a 2-layer convolutional model for computing activation features. For OMNIGLOT we used a 3-layer, 2-head Transformer and for TIEREDIMAGENET, we used a simplified 1-layer Transformer with 8 heads. In all our experiments, we trained the network on a single GPU for $4M$ steps with SGD with an exponential LR decay over $100\,000$ steps with a decay rate of $0.97$. We noticed some stability issues when increasing the number of tasks and had to decrease the learning rate to compensate: for OMNIGLOT experiments, we used a learning rate $10^{-4}$ for up to 4 tasks and $5\times10^{-5}$ for 5 tasks. For TIEREDIMAGENET, we used the same learning rate of $5\times10^{-6}$ for training with any number of tasks $T$. We trained the CHT models with the class-incremental objective (6), but evaluated them for both task-incremental and class-incremental scenarios.

---

[2]In contrast with cross-entropy, we do not need to have the head layer dimension to be equal to the number of predicted labels when using the Prototypical Loss.

## 6.1 Learning from mini-batches

We first consider a case where every task includes the same set of classes. Specifically, we compared the following three models using a set of four 5-way 1-shot support set batches $S^{(1)}, \ldots, S^{(4)}$ that consist of the same set of classes from TIEREDIMAGENET:

$$
\begin{aligned}
\theta^{(a)} &\equiv a_\psi(S^{(1)} + S^{(2)} + S^{(3)} + S^{(4)}, \theta_0), \\
\theta^{(b)} &\equiv a_\psi(S^{(3)} + S^{(4)}, a_\psi(S^{(1)} + S^{(2)}, \theta_0)), \\
\theta^{(c)} &\equiv a_\psi(S^{(4)}, a_\psi(S^{(3)}, a_\psi(S^{(2)}, a_\psi(S^{(1)}, \theta_0)))),
\end{aligned}
$$

where $+$ operation denotes a simple concatenation of different support set batches. For this experiment, we used the cross-entropy loss (since the label set was the same for all $S^{(i)}$) and each support set batch $S^{(i)}$ contained a single example per class. We observed that the test accuracies for $\theta^{(a)}$, $\theta^{(b)}$ and $\theta^{(c)}$ were equal to 67.9%, 68.0% and 68.3% respectively, all within the statistical error range ($\pm 0.4\%$). At the same time, HT trained with just $S^{(1)}$ or $S^{(1)} + S^{(2)}$ (with 1 or 2 samples per class respectively) performed significantly worse, reaching the test accuracies of 56.2% and 62.9% respectively. This demonstrates that the proposed mechanism of updating generated CNN weights using information from multiple support set batches can achieve performance comparable to processing all samples in a single pass with HT.

## 6.2 Learning from tasks within a single domain

Next, we consider a scenario where the tasks consist of classes drawn from a single overall distribution. We present the results of two models: one trained on 20-way, 1-shot tasks with classes sampled from OMNIGLOT dataset, and anther trained on 5-way, 5-shot tasks with classes sampled from TIEREDIMAGENET dataset.

We compare the performance of CHT to two baseline models. The first is a CONSTANT PROTONET (CONSTPN), which represents a vanilla Prototypical Network, as described in Snell et al. (2017). In this approach, a universal fixed CNN network is trained on episodes from $\mathcal{C}_{\text{train}}$. This constant network can be applied to every task separately by projecting the support set as prototypes for that task and computing the prediction with respect to these prototypes. Strictly speaking, this is not a continual learning method, since it treats every task independently and has no memory of previous tasks. For the best results on this baseline, we had to increase the number of classes by a factor of 5 during training (e.g. for 20-way OMNIGLOT evaluation we have trained it with 100-way problems).

The second baseline we used specifically for the class-incremental learning is a MERGED HYPERTRANSFORMER (MERGEDHT), where we combine all the tasks and train a single original HT instance as a single task. This method does not solve a continual learning problem, since it has the information about all the tasks from the beginning, but it produces a solution for every class and task that we can still be compared to the weights generated by the CHT.

Each trained model is applied to both task-incremental (Figure 3) and class-incremental (Figure 4) settings. To understand the effect of continual learning with multiple tasks, each column represents a separate run of the CHT trained on $T = 2, 3, 4$ or 5 tasks in total (for training a higher $T$, see the results in the Appendix). To demonstrate the recurrence of the method, we extended the number of tasks to 5 for the evaluation regardless of how many tasks it was trained on. Each plot shows 5 curves corresponding to the CHT, split into two groups: bullet marker ($\bullet$) for tasks that the model was trained for and diamond marker ($\diamond$) for extrapolation to more tasks.

**Task-incremental learning.** We start by analysing the task-incremental learning results. For the OMNIGLOT dataset, we saw no signs of catastrophic forgetting for the CHT. In fact, we observed a positive backward knowledge transfer, where the performance on past tasks *improved* as more weights were generated. For example, in most cases, the performance of $\theta_1$ (green markers) was higher than $\theta_0$ (orange markers), and $\theta_2$ was higher than both $\theta_1$ and $\theta_0$. Additionally, as the number of tasks increased, the overall performance of the CHT also increased, with the model trained on $T = 5$ tasks performing better than the one trained on $T = 2$ tasks.

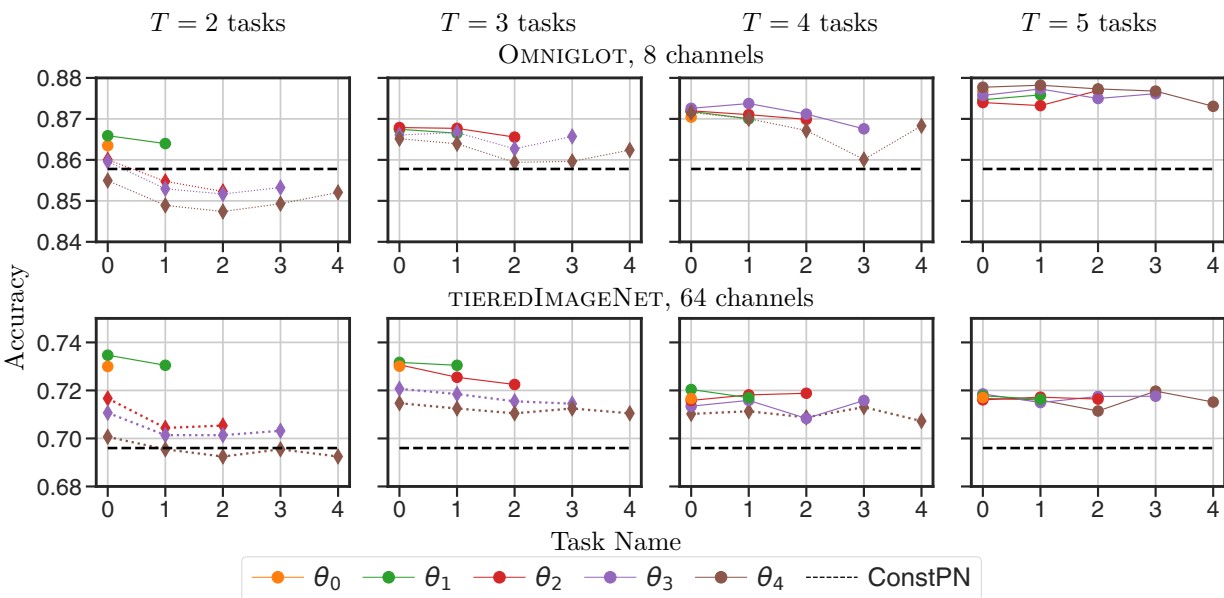

Figure 3: Task-incremental learning on OMNIGLOT and TIEREDIMAGENET. Each column represents a different CHT trained with a total of $T = 2, 3, 4$ or $5$ tasks. The tasks marked with a bullet symbol ($\bullet$) correspond to the terms in the objective function (3) that are being minimized. The lines marked with the diamond symbol ($\diamond$) show the extrapolation of the trained CHT to a larger number of tasks. The confidence intervals do not exceed 0.5%.

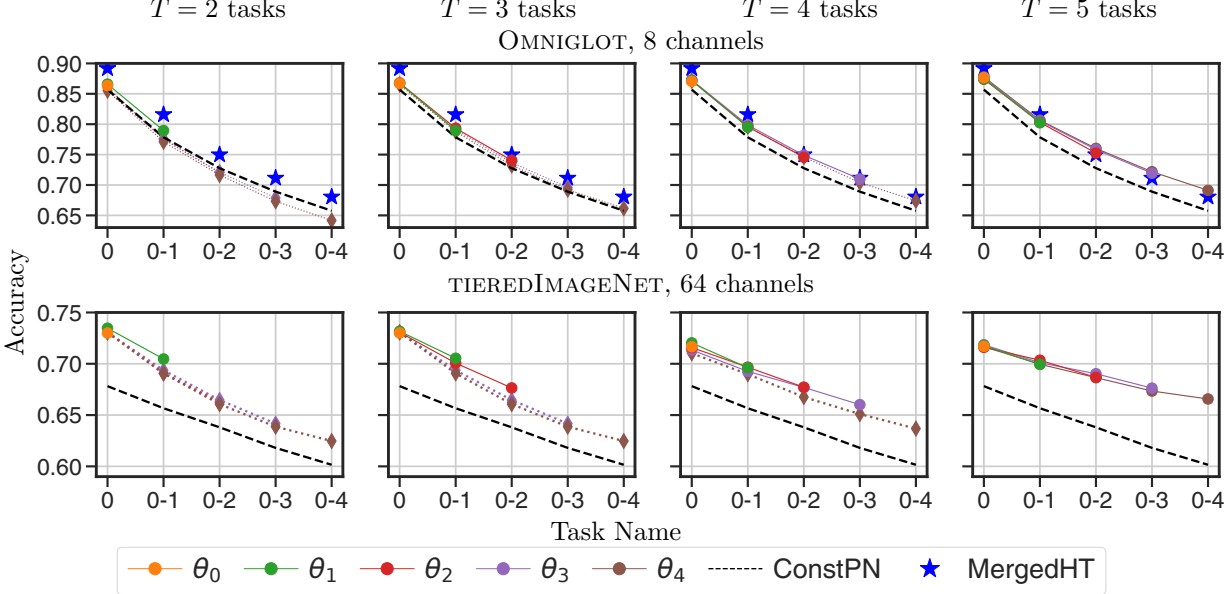

Figure 4: Class-incremental learning on OMNIGLOT and TIEREDIMAGENET. Each column represents a different CHT trained with a total of $T = 2, 3, 4$ or $5$ tasks. The tasks marked with a bullet symbol ($\bullet$) correspond to the terms in the objective function (3) that are being minimized. The lines marked with the diamond symbol ($\diamond$) show the extrapolation of the trained CHT to a larger number of tasks. The confidence intervals do not exceed 0.5%.

Table 1: Performance comparison of different methods for learning up to 5 tasks on OMNIGLOT dataset.

| Method | Task name | | | | |
|---|---|---|---|---|---|
| | 0 | 0-1 | 0-2 | 0-3 | 0-4 |
| Pretraining | 38.4 | 21.1 | 8.3 | 4.8 | 3.9 |
| MAML++ | 81.4 | 58.2 | 33.5 | 24.4 | 19.8 |
| EWC | 33.4 | 20.4 | 9.0 | 4.7 | 3.7 |
| Continual HT (ours) | 87.2 | 80.0 | 75.1 | 76.8 | 69.3 |

For the TIEREDIMAGENET dataset, the results were better than the CONSTPN baseline, but the positive backward knowledge effect effect was not as pronounced as it was for the OMNIGLOT dataset. The performance for every training task remained roughly the same for all generated weights, indicating that the model did not suffer from catastrophic forgetting.

Overall, the CHT consistently outperformed the CONSTPN baseline, particularly when applied to the same or lower number of tasks it was trained on. Although the accuracy of the CHT did decrease slightly when it was applied to more tasks than it was trained on, this decrease was not significant. In fact, even when CHT was trained on only $T = 3$ tasks, generating weights for one of two additional tasks still resulted in better performance than the CONSTPN baseline.

**Class-incremental learning.** In the class-incremental learning setting, the task name is given by two numbers indicating the range of tasks we used for evaluation (e.g. task name 0-3 corresponds to four tasks from 0 to 3). The black constant dashed line is the baseline performance of the CONSTPN, which uses a fixed embedding and does not differentiate between tasks. The starred blue markers represent a separate run of the HT for a particular configuration of merged tasks.

As one can see in the Figure 4, the accuracy of all the models decreased as more tasks were included in the prediction. This was expected because the size of the generated CNN did not change, but the number of classes that needs to be predicted was increasing. For OMNIGLOT dataset we again saw the positive backwards transfer taking place, with CHT models trained on more tasks $T$ performing better overall. For a given model trained on a fixed $T$, the performance was comparable. This demonstrates the preemptive property of the CHT, where models trained for a certain number of tasks can still be run for any smaller number of tasks with similar performance.

When comparing the results to the baselines, the CHT had better results than the CONSTPN up to the number of tasks $T$ it was trained for, and the extrapolation results improved as $T$ increases. Interestingly, for the case of $T = 5$ the CHT was able to outperform even the MERGEDHT baseline for the OMNIGLOT, even though the MERGEDHT had access to information about all tasks from the beginning. This suggests that having more classes to classify makes the learning problem difficult for the original HT, as the image embeddings may not be able to learn good embeddings. This is particularly evident in the TIEREDIMAGENET dataset, where the performance of the MERGEDHT is so low that it falls below 60%, even for the 0-1 task.

We have also compared the results of CHT with other, more traditional baselines from few-shot learning and continual learning literature. Specifically, we consider the following methods:
- *Pretraining.* Pretraining a network on a full training set, then fine-tuning it on an input sequence of tasks from the test set.
- *MAML++.* An example of a Few-Shot Learning method (Antoniou et al., 2019).
- *EWC.* An example of a Continual Learning method, we run EWC (Kirkpatrick et al., 2017) on a pretrained network from above.

Table 1 shows the results. Notice that MAML++ performed well on the first task, but not able to retain the knowledge for subsequent tasks, because of catastrophic forgetting. Pretraining methods and EWC were not able to learn sufficiently well even the first task, because they were not able to learn from few-examples.

### 6.3 Learning from tasks across multiple domains

In the experiments described above, the support and query sets for each task were drawn from the same general distribution, and the image domain remained consistent across all tasks. If the tasks were drawn

from different distributions and different image domains, we would expect task-agnostic ConstPN approach to suffer in accuracy because it would need to find a universal representation that works well across all image domains. In contrast, the CHT approach could adapt its sample representations differently for different detected image domains, leading to improved performance.

We verify this by creating a multi-domain episode generator that includes tasks from various image datasets: Omniglot, Caltech101, CaltechBirds2011, Cars196, OxfordFlowers102 and StanfordDogs. We compared the accuracy of the ConstPN and CHT on this generator using episodes containing two tasks with 5-way, 1-shot problems. The generated CNN model had 16 channels with 32 channels for the final layer. Other parameters were the same as those used in the tieredImageNet experiments. The ConstPN achieved the accuracy of 53% for task 0, 52.8% for task 1 and 50.8% for combined tasks. The CHT achieved the accuracy of 56.2% for task 0, 55.2% for task 1 and 53.8% for combined tasks. The accuracy gap of nearly 3% between these two methods, which is larger than the gap observed in the Omniglot and tieredImageNet experiments, suggests that the CHT is better at adapting to a multi-domain task distribution.

## 7 Discussion

While the computational cost of training the hypernetwork can be quite large, the training process is a one-time investment on a meta-training distribution of tasks. Once trained, the HyperTransformer can generate target CNN architectures for new sets of tasks in a matter of seconds (e.g., 10 seconds as demonstrated in our experiments). This efficiency in generating new architectures is a key advantage when operating in dynamic task environments.

Since HyperTransformer requires a separate Transformer network in order to generate a target CNN, whose architecture and optimization parameters can be considered hyper-parameters of our method. However, in scenarios with novel datasets, a practitioner may need to explore hyperparameter tuning to optimize performance. A separate sensitivity analysis needs to be done to understand the role of various hyperparatemeters of a generative Transformer network.

## 8 Conclusions

The proposed Continual HyperTransformer model has several attractive features. As an efficient few-shot learner, it can generate CNN weights on the fly with no training required, using only a small set of labeled examples. As a continual learner, it is able to update the weights with information from new tasks by iteratively passing them through HT. Empirically, we have shown that the learning occurs without catastrophic forgetting and may even result in positive backward transfer. By modifying the loss function from cross-entropy to the prototype loss, we defined a learning procedure that optimizes the location of the prototypes of all the classes of every task. A single trained CHT model can be used in both task-incremental and class-incremental scenarios.

## 9 Broader Impact Statement

This paper is focused on developing new methods for continual few-shot learning. We do not envision these methods being used for harmful purposes more so than other algorithms in a class of few-shot or continual learning.

The authors of this paper are committed to making sure that the methods we develop are used with safety concerns in mind. We believe that continual few-shot learning has the potential to be a powerful tool for good, but it is important to use it responsibly. While the technology itself does not directly facilitate harm to living beings or raise immediate safety concerns, its broader impact needs to be carefully considered in light of potential ethical, societal, and environmental implications. We are committed to working with the research community to mitigate potential risks. We welcome any feedback from the reader regarding a potential misuse of the methods described in the paper that we did not describe in this section.

## 10    Acknowledgements

The authors would like to thank Nolan Miller, Gus Kristiansen, Jascha Sohl-Dickstein and Johannes von Oswald for their valuable insights and feedback throughout the project.

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

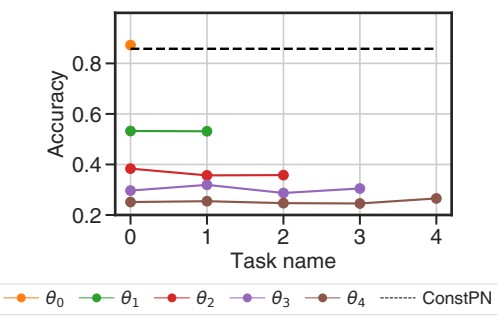

Figure 5: The accuracy of the HT model trained for $T = 5$ using the cross-entropy loss. The accuracy of the first weight $\theta_0$ is high and is better than the accuracy of the CONSTPN model's embeddings. However, when more tasks are added, the accuracy drops dramatically due to collisions between the same classes for different tasks in the cross-entropy loss.

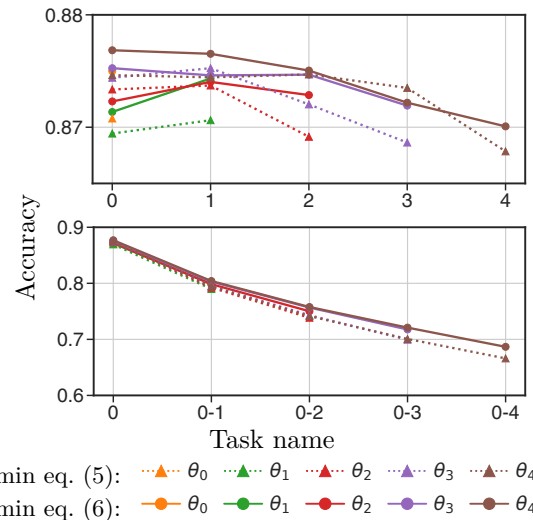

Figure 6: CHT trained using task-incremental objective (5) vs. class-incremental objective (6).

## A    Additional experiments and figures

### A.1    Learning with cross-entropy loss

Figure 5 shows the results of an attempt to do learn multiple tasks using a HT with a cross-entropy loss. Since the size of the last layer's embedding is not increased, the model can only predict the class labels within the task and not the task themselves, which corresponds to the domain-incremental learning setup. Additionally, the same class from different tasks are mapped to the same location in the embedding space, leading to collisions when more tasks are added. This is why the accuracy drops significantly as the number of tasks increases. On the other hand, CONSTPN model is more flexible because the prototypes for each task are computed from the support set of that task and do not have to be fixed to a one-hot vector as in the cross-entropy loss.

### A.2    Training using task-incremental and class-incremental objectives

Figure 6 compares the accuracy of two different models trained with task-incremental (using equation (5)) and class-incremental (using equation (6)) objectives. The performance of both models on task-incremental problems are similar, while the model trained with the class-incremental objective performs better on class-incremental problems.

### A.3    Analysis of prototypical embeddings using UMAP

To better understand the quality of the learned prototypes, we conducted an experiment on a TIEREDIMA-GENET 5-way, 5-shot problem. We selected a random episode of 5 tasks and ran them through the model, producing weights $\theta_0$ to $\theta_4$ along with the prototypes for each class of every task. We then computed the logits of the query sets, which consisted of 20 samples per class for each task. The resulting 40-dim embeddings for the prototypes and query sets were concatenated and projected onto a 2D space using UMAP (Figure 7). Note that the prototypes from the earlier tasks remain unchanged for the later task, but their 2D UMAP projections are different, because UMAP is a non-parametric method and it must be re-run for every new $\theta_k$. We tried our best to align the embedding using the Procrustes alignment method.

The plot shows that the embeddings of the tasks are well separated in the logits space, which helps explain why the model performs well for both task- and class-incremental learning. Normalizing the softmax over the classes within the same tasks or across all tasks made little difference when the tasks are so far away from each other. On the right of Figure 7, we show the projection of the CONSTPN embedding of the same

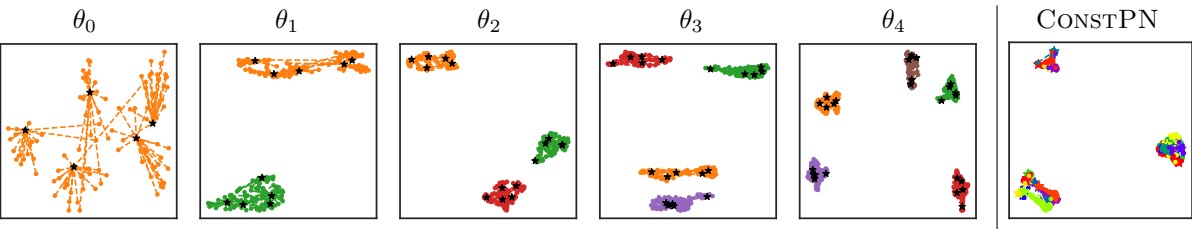

Figure 7: *Left 5 plots:* the UMAP projection of the CHT prototypes and the query set embeddings for different generated weights, where the points are colored based on the task information. The query set points are connected with their corresponding prototypes using dashed lines. *Right plot:* UMAP projection of the CONSTPN embedding for 25 different classes from TIEREDIMAGENET. Embeddings are aligned using Procrustes alignment.

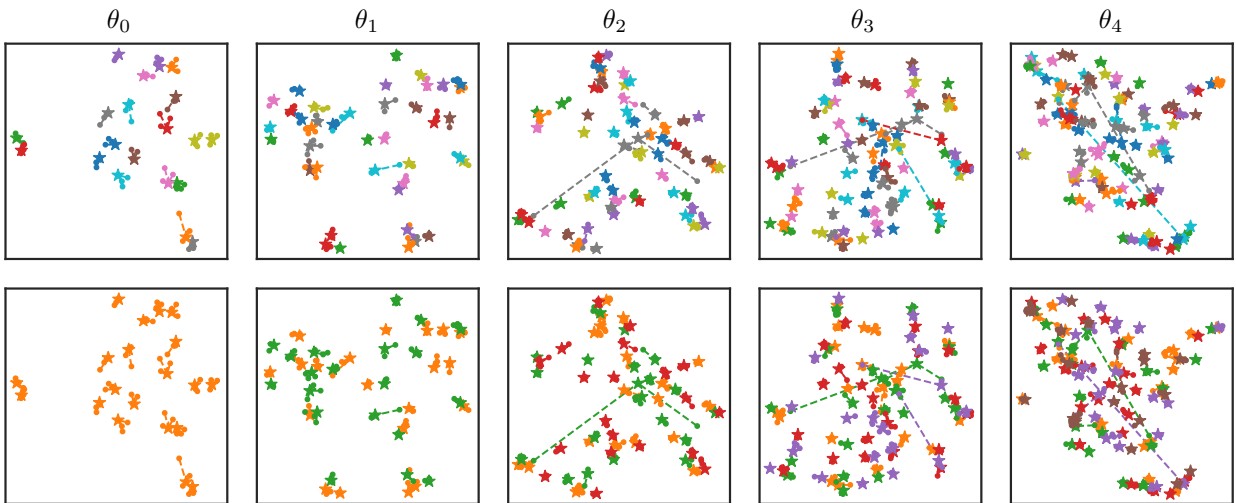

Figure 8: The UMAP projections of 20-dimensional embeddings of the prototypes and query set for different weights obtained from incremental HT training. The query set points are connected to their corresponding prototypes using dashed lines. In the top plot, the points are colored according to their class information, while in the bottom plot they are colored according to their task information. Embeddings are aligned using Procrustes alignment.

25 classes. The CONSTPN model does not make a distinction between tasks and treats each class separately. The fact that 3 clusters emerge has to do purely with the semantics of the chosen clusters and the way the CONSTPN model groups them. This also helps to explain why the CHT model performs better than the CONSTPN, as it separates the tasks before separating the classes within each task.

The UMAP embedding for the OMNIGLOT dataset using ProtoNet (Figure 8) appears to be different from similar embedding projection of TIEREDIMAGENET dataset. In particular, the embeddings from different tasks seem to overlap, while in the TIEREDIMAGENET embedding they are separated. This may be due to the fact that the classes in the OMNIGLOT dataset are more closely connected than those in the TIEREDIMAGENET dataset. Interestingly, despite the overlap between the classes from different tasks, the final accuracy is still high and only slightly degrades as more tasks are added.

## A.4 Learning with more tasks

Our analysis primarily focused on the performance of the CHT on up to 5 tasks. However, as shown in Figure 9 the CHT model is capable of handling a much larger number of tasks $T$. Similar to the results in

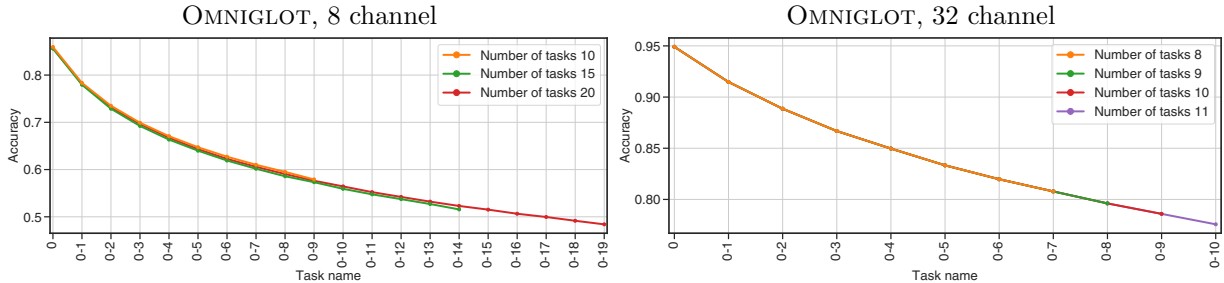

Figure 9: OMNIGLOT with 8 or 32 channels trained with a different number of tasks $T$.

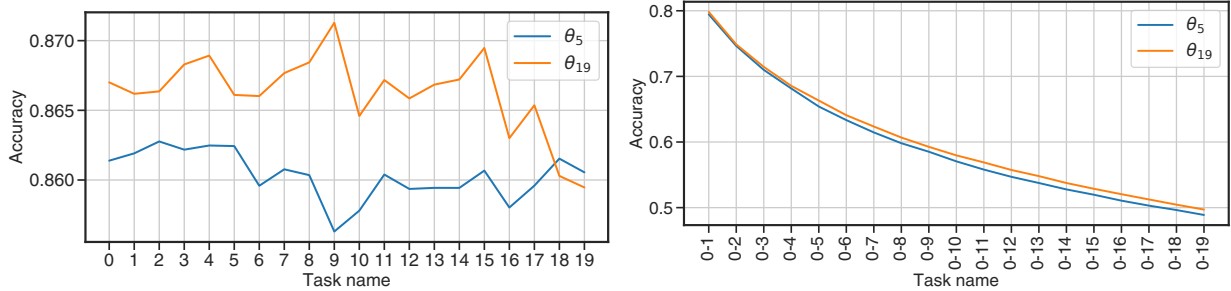

Figure 10: OMNIGLOT with 8 channels trained for $T = 20$ tasks. Here we show the final weight $\theta_{19}$ generated along with an intermediate $\theta_5$ for task-incremental (*left*) and class-incremental (*right*) learning.

the main text, the nearly overlapping curves in the graph indicate that the model trained for $T$ tasks can maintain the same level of accuracy when applied to a larger number of tasks.

Figure 10 shows 8-channel OMNIGLOT evaluated on task-incremental and class-incremental objectives.

### A.5 Continual HyperTransformer vs MergedHT for tieredImageNet

Figure 11 shows a zoomed out view of the results presented in Figure 4). It illustrates the significant difference in performance between the MERGEDHT and the CHT models.

### A.6 Additional figures for Omniglot task-incremental and class-incremental learning

Figures 12, 13, 14 and 15 show additional experiments with the OMNIGLOT dataset using different number of channels in the CNN.

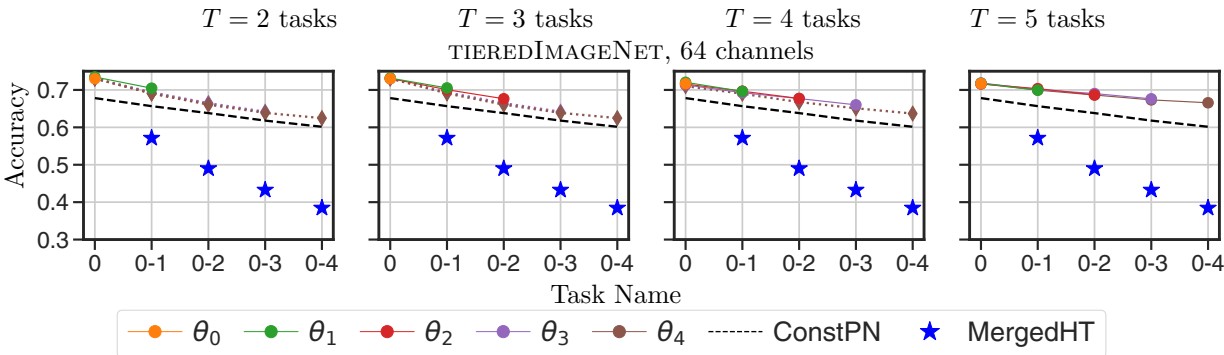

Figure 11: Zoomed out view of Figure 4 so that the results of the MERGEDHT is visible.

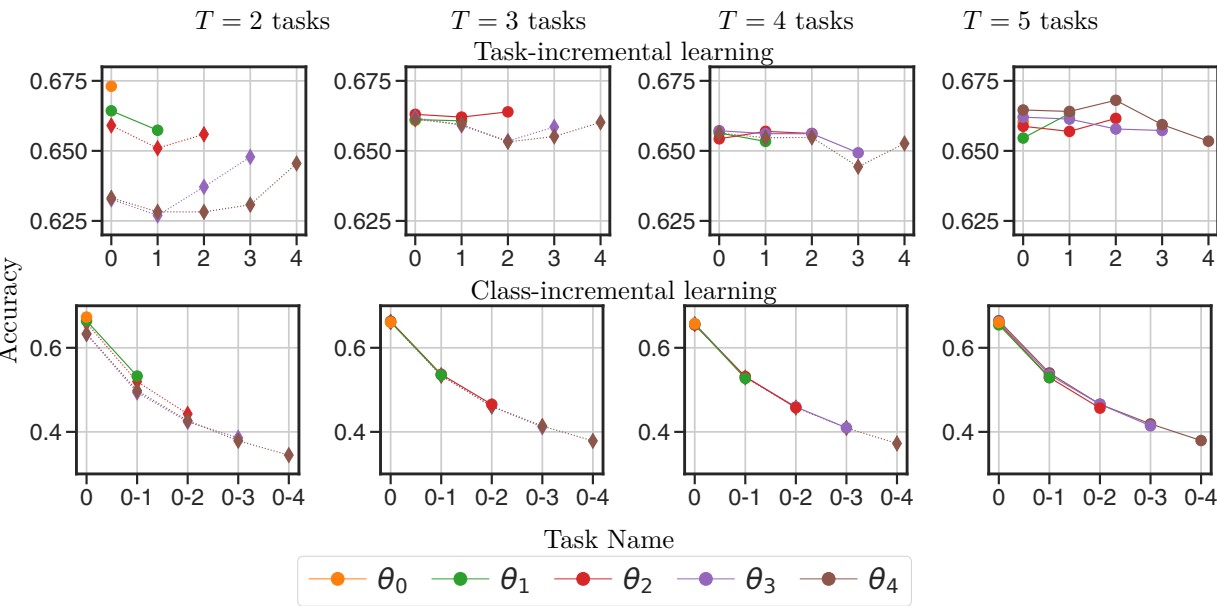

Figure 12: Task-incremental and class-incremental learning on the OMNIGLOT dataset with 4-channels convolutions.

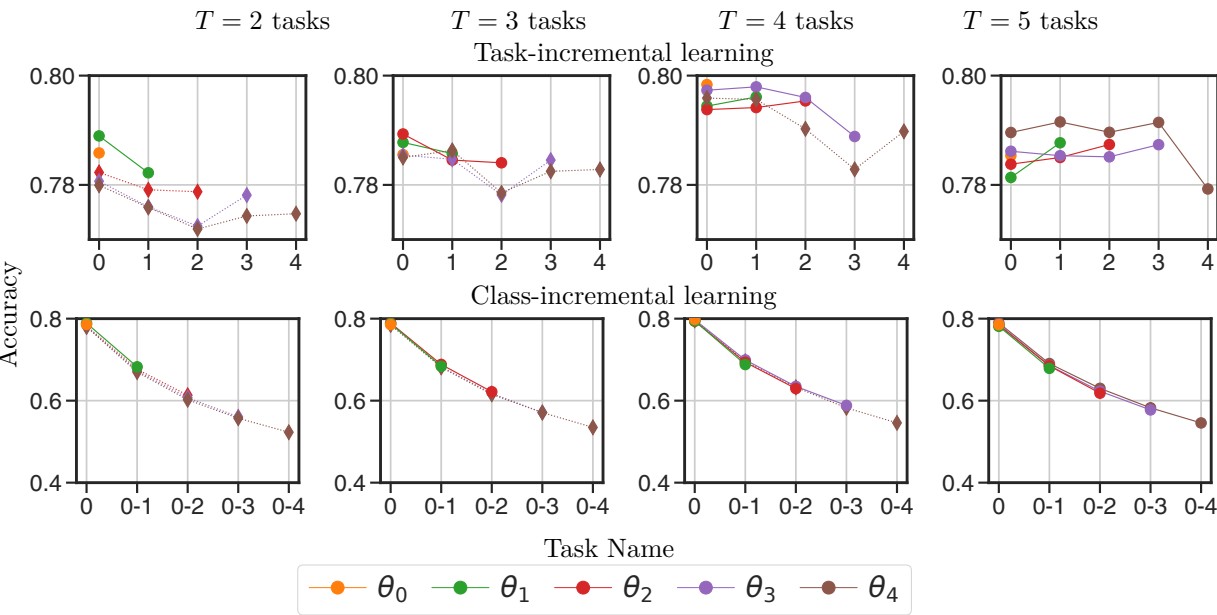

Figure 13: Task-incremental and class-incremental learning on the OMNIGLOT dataset with 6-channels convolutions.

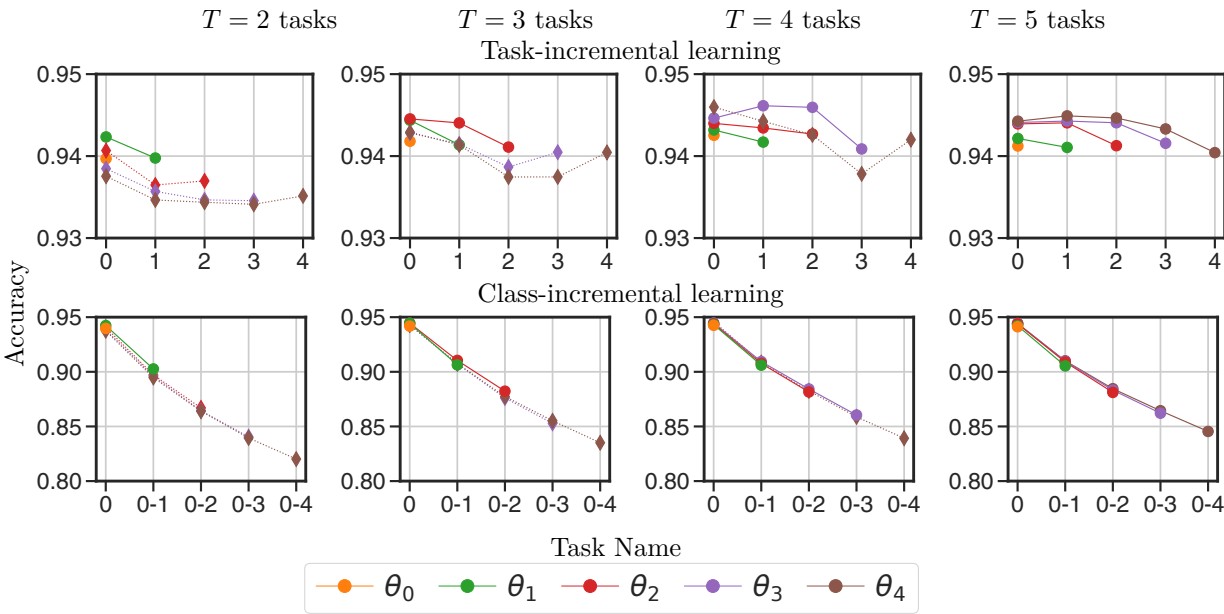

Figure 14: Task-incremental and class-incremental learning on the OMNIGLOT dataset with 16-channels convolutions.

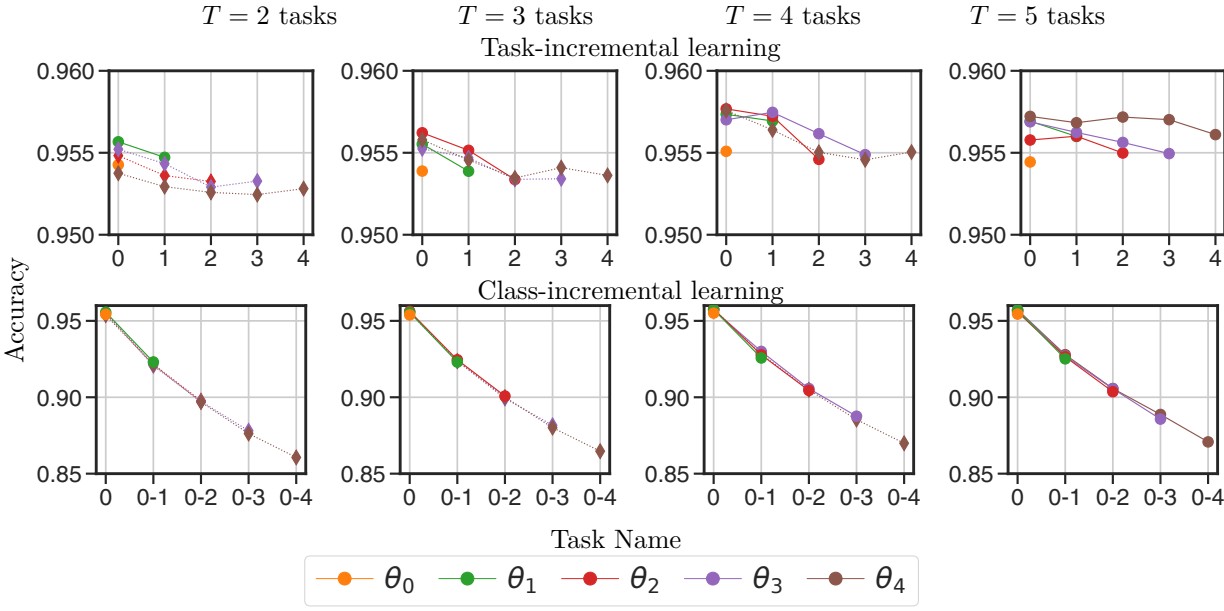

Figure 15: Task-incremental and class-incremental learning on the OMNIGLOT dataset with 32-channels convolutions.

