# OpenReview forum: "Continual HyperTransformer: A Meta-Learner for Continual Few-Shot Learning"
_TMLR — Accepted by TMLR_

### Review · Reviewer_frdT · 2023-11-11

**Summary Of Contributions:**

The paper proposes a variant of HyperTransformer (HT) originally developed for few-show learning, Continual HyperTransformer  (CHT) for the task of continual few-show learning. By feeding previously generated weights into CHT, the generated weights contains information from a new task while retaining the knowledge about tasks have already been seen. With the prototypical loss, CHT achieves competitive results on two standard benchmarks, Omniglot and tieredImageNet, in several continual few-shot learning scenarios.

**Audience:**

Yes

**Broader Impact Concerns:**

The authors adequately addressed the ethical implications of the work.

**Claims And Evidence:**

Yes

**Requested Changes:**

- Please describe the architecture of CHT in more detail.
- Please add more baselines, if possible. Comparing with only a single baseline is not enough for verifying the superiority of the proposed method (MergedHT is used only for class-incremental learning on Omniglot).

**Strengths And Weaknesses:**

Strengths

+ Overall, the various scenarios of continual few-shot learning that the paper covers, related work and all experimental setups are presented clearly.
+ Extensive experiments showcase the validity of the proposed method, CHT.

Weaknesses
- The authors should describe the architecture of CHT in more detail. Figure 2 alone is not sufficient for explaining the model architecture.

The well-designed experiments explore various aspects of CHT in the target problem, continual few-shot learning, which clearly shows the effectiveness of the method. It seems that this work is quite incremental because it adapts the existing model, HT, to the continual learning and does not introduce any new components in the training objective; it just uses the existing prototypical loss. However, I don't believe that this should not be a reason for rejection by itself, according to the acceptance criteria of the journal.

---

> ### Author Response · Authors · 2024-01-19
> **Reply to Reviewer frdT**
>
> We thank the reviewer for taking time to read and evaluate our submission. We are glad the reviewer noted that our submission covers various scenarios of continual FSL and the experiments are extensive and demonstrate the validity of our proposed approach.
>
> > The authors should describe the architecture of CHT in more detail. Figure 2 alone is not sufficient for explaining the model architecture.
>
> Thanks for the feedback. This is a valid point. We have worked on expanding the description of the model’s structure. Specifically, we have added more information on HyperTransformer architecture and how exactly the CNN weights are generated.
>
> > It seems that this work is quite incremental because it adapts the existing model, HT, to the continual learning and does not introduce any new components in the training objective:
>
> While our approach involves adapting the existing HT model for continual learning (CL) without introducing entirely new components, we emphasize that our modifications are purposeful and impactful. We believe that the proposed modifications to HT model are not incremental, but rather _organic_ and that the original model is actually well-suited for the CL. This connection to the CL was not obvious in the original HT submission.
>
> There are several non-trivial and novel modifications that we made:
> - We pass previously generated weights of one task as the input to the HT handling another task. This is not obvious given that these weights actually play a dual function: they serve as the embedding encoding of the previously observed tasks as well as the CNN weights to solve this task.
> - The Prototypical Loss was originally used in the context of few-shot learning (FSL). We propose to use it in a multi-task CL, since it allows for almost trivial formulation of both class-incremental and task-incremental CL objectives.
> - We connect Prototypical Loss to MAML (Section 5), situating our approach within the broader metalearning literature. This provides meaningful theoretical grounding for our modifications.
>
> In addition, in the experimental section we have demonstrated a series of non-trivial and, what we think, exciting results, such as:
> - works in continual learning scenario as well as if the data is provided all at once (Section 6.1)
> - does not suffer from catastrophic forgetting for up to 5 tasks and works in both task-incremental and class-incremental settings (Section 6.2).
> - in some cases, we are able to demonstrate highly desirable, but very elusive positive backward transfer.
>
> We believe these modifications and strong experimental results make our work interesting and exciting for the reader.
>
> > MergedHT is used only for class-incremental learning on Omniglot.
>
> MergedHT is a baseline that merges all the tasks together in a single large task, essentially solving a single $KT$-way problem, where $T$  is a number of tasks and $K$  is the number of classes in each task.
> It is not applicable for task-incremental learning (Fig.3) since there we have an additional information of knowing a task attribute and thus the problem is to solve a specific task, not all tasks together.
> For class-incremental learning (Fig.4), we need to solve for both task and class attributes and MergedHT is a fitting baseline. We didn’t show it in Fig.4, since it is so bad that it fell off the charts (we alluded to it at the end of section 6.2). Figure 11 showed the zoomed out view of Fig.4 with MergedHT visible. We will clarify these points in the main text of the paper.

---

### Review · Reviewer_nY3Y · 2024-01-03

**Summary Of Contributions:**

This work focuses on continual few-shot learning, whereby a model learns over a stream of data and tasks while having access to few data points at the time and no access to past data. An adaptation of an existing model, HyperTransformers, is proposed called Continual HyperTransformers. The adaptation is straightforward, conditioning the model on past tasks, specifically the generated weights from past tasks. The paper claims the model exhibits limited forgetting and shows evidence of the opposite, called 'positive backward transfer', that is that the model gets better with new tasks.

**Audience:**

Yes

**Broader Impact Concerns:**

Ok.

**Claims And Evidence:**

Yes

**Requested Changes:**

See above.

**Strengths And Weaknesses:**

Strengths
- Overall, this is an ok paper, with a modest technical contribution and ok empirical justification. The proposed model is an adaptation of an existing one that works well with few-shot learning, that is HyperTransformers. Empirical validation shows that the continual version of it can also handle streaming settings.
- The related work is broadly covered with respect to neighbor fields, including few-shot learning, continual learning, etc.
- In that sense, this paper also extends existing works towards a novel data setting, where not only the model is exposed to limited data (aka few-shot learning) but is done in a continual way also.

Weaknesses
- The paper is not as exciting, in that it seems an existing method and an existing data setting are rather straightforwardly extended and results are decent but unsurprising. I understand that this is a comment that is hard to argue for or against, my point is that at the moment there seems to be no strong conclusion or insight from the work. Indeed, adapting an existing model towards a specific new (although not unheard of) direction gives the expected results. In that sense, the novelty is on the low side.
- I think the comparisons, and general positioning is lacking, which is also evident in that the paper makes only 'internal' comparisons but does not compare much against other published baselines. For instance, I know that T. Tuytelaars from KU Leuven has worked extensively on continual learning as well as few-shot learning, including also a survey on the subject matter (and I am sure there must be more baselines on such a broad topic). This is missing at the moment, and I would advise looking at the related literature before having a more extensive review
https://scholar.google.be/citations?hl=en&user=EuFF9kUAAAAJ&view_op=list_works&sortby=pubdate
https://scholar.google.be/citations?view_op=view_citation&hl=en&user=EuFF9kUAAAAJ&citation_for_view=EuFF9kUAAAAJ:isU91gLudPYC

Some more works with simple Google search
https://aclanthology.org/2021.emnlp-main.460/
https://dl.acm.org/doi/abs/10.1145/3543507.3583262
https://github.com/huang50213/AIM-Fewshot-Continual

- I find the text clear but the overall composition and description of the method lacking sufficient detail. For instance, it is not clear what is the definition of tasks here, as it is not explained really, not even in the experiments section. One could think of as tasks to be new types of classes (but is this then a 'new task' or simply extending the class set)? A more natural definition would be having classification, localization, segmentation, detection etc as different tasks. Either way, right now it is not easy to understand what exactly the method does and whether the empirical validation is sufficient. More details in random order
-- Why is positive backward transfer working? Is there any theoretical or intuituve justification?
-- In general, the whole claim is rather ad hoc, which is I guess anyway expected, as it is very hard to prove theoretically that an online algorithm would generalize. However, I feel a bit uneasy confirming that this algorithm would necessarily work well on a new dataset or a new set of tasks.
-- The CNN weights are generated, thus the CNN has no training itself. Rather, the training is on the transformer that generates the CNN parameters on the fly. If my understanding is correct, does one really need a batch norm layer which is mostly for training? I suppose one would still need it for practical purposes, or because even 'indirect' training would require some form of gradient normalization, however, I would expect some clarity there.
-- On a similar matter, if the classifier is determined dynamically as in equations 4-6 with prototytpes, is there potentially a problem of scale, when adding many new classes, thus creating gradients of much larger magnitudes?
-- What exactly is the point of section 5? Does it give any new insight from MAML literature or is it more of a 'hey look what i found out' highlight?
-- How important is the order with which the classes and tasks are presented. Especially when not having a recurrent setting with shared weights, wouldn't this affect results significantly? I would expect at least an ablation study here.

In general, I think that this is a very hard task that the paper is addressing and I want to express my appreciation for the difficulty.

---

> ### Author Response · Authors · 2024-01-19
> **Reply to reviewer nY3Y, part 1**
>
> We thank the reviewer for carefully reading our manuscript and providing very useful feedback. We have tried to answer all the questions that the reviewer has posed.
>
> > no strong conclusion or insight from the work
>
> We believe that there are several interesting insights from out work:
> - our continual learning method that does not rely on replay buffers, weight regularization, or architectural adaptations. The continual learning happens organically inside the HyperTransformer with no special treatment to prevent catastrophic forgetting.
> - our method is flexible and can handle different learning scenarios: mini-batch learning, task- and class-incremental continual learning. This demonstrates versatility of the proposed approach.
> - we have demonstrated that, within the scope of our experiments and datasets, the proposed method does not suffer from catastrophic forgetting and, surprisingly, even improves the results on already learned tasks when presented with new tasks (positive forward transfer). This confirms the soundness of the approach as a valid method for continual few-shot learning.
>
> > Missing literature
>
> We thank the reviewer for pointing out some of the missing literature. We will gladly add them to the literature review. Specifically, A continual learning survey mentioned by the reviewer gives an excellent overview of the continual learning methods. However, it does not cover any methods that are applicable specifically to a few-shot learning setting. Methods do need to be adapted for this scenario, since there are only few samples
>
> Pasunuru et al [1] doesn’t define a new method, but rather a benchmark suite for continual learning specifically for NLP problems.
>
> Wang et al [2] proposes a heterogeneous framework that uses visual as well as additional semantic textual concepts. In our paper we only focus on visual input for model training and predictions.
>
> Finally, we actually do cite Lee et al [3] in our manuscript as an example of related, but different class of Incremental few-shot learning, where the goal is to update the existing base classifier with few samples of novel classes. In contrast, in our paper, we learn a classifier from scratch for a sequence of user-provided tasks.
>
> [1] Pasunuru, Ramakanth, Veselin Stoyanov, and Mohit Bansal. “Continual Few-Shot Learning for Text Classification”
> [2] Wang, Xin, Yue Liu, Jiapei Fan, Weigao Wen, Hui Xue, and Wenwu Zhu. “Continual Few-shot Learning with Transformer Adaptation and Knowledge Regularization”
> [3] Lee, Eugene, Cheng-Han Huang, and Chen-Yi Lee“Few-Shot and Continual Learning with Attentive Independent Mechanisms”
>
>
> > it is not clear what is the definition of tasks here, as it is not explained really, not even in the experiments section. One could think of as tasks to be new types of classes (but is this then a 'new task' or simply extending the class set)? A more natural definition would be having classification, localization, segmentation, detection etc as different tasks. Either way, right now it is not easy to understand what exactly the method does and whether the empirical validation is sufficient.
>
> In the beginning of Section 3, we define precisely what we mean by Tasks in our paper. Each task consists of a K-way, N-shot support set and K-way, $\hat N$-shot query set. We also discuss several options on whether each task consists of the same classes (aka mini-batch learning evaluated in section 6.1), different classes within the same domain (aka lifelong learning evaluated in section 6.2) or different classes from different domains (evaluated in section 6.3). We believe that, within the scope of the few-shot learning classification problem, these three options cover most of the interesting practical possibilities.
>
> Having heterogeneous tasks that correspond to different loss functions altogether (such as classification, localization etc) is a very interesting idea! Our method would support this, since our final loss function (3) is just a sum of individual losses for each task, but it would probably require at the very least modifying the loss to use the weighted sum of different heterogeneous losses. We leave this as an exciting future work direction.

---

> > ### Author Response · Authors · 2024-01-19
> > **Reply to reviewer nY3Y, part 2**
> >
> > > Why is positive backward transfer working? Is there any theoretical or intuitive justification?
> >
> > We believe that the positive backward transfer happens for two reasons:
> > - When the classes in different tasks are closely related (for example, different cat breeds), learning about one class can benefit the model's understanding of previously learned, similar classes.
> > - more importantly, since each task consists of only a few examples, every single example carries significant weight. In the extreme scenario of only one example per class per task, learning from that example can positively impact the model's knowledge of related classes learned earlier, as each sample holds a high density of information.
> >
> > Positive backward transfer is less likely to occur for tasks with classes from different domains or when the number of examples per class is sufficient.
> >
> > > In general, the whole claim is rather ad hoc, which is I guess anyway expected, as it is very hard to prove theoretically that an online algorithm would generalize. However, I feel a bit uneasy confirming that this algorithm would necessarily work well on a new dataset or a new set of tasks.
> >
> > Generally, it is quite hard to say what would happen for a new novel dataset or a set of tasks, that is why we rely on extensive empirical evaluations. Intuitively, the learning happens because of meta-generalization. In other words, if the distribution of tasks during meta-training matches the distribution during the meta-test, there might be a reasonable expectation that the generalization will work.
> >
> > > The CNN weights are generated, thus the CNN has no training itself. Rather, the training is on the transformer that generates the CNN parameters on the fly. If my understanding is correct, does one really need a batch norm layer which is mostly for training? I suppose one would still need it for practical purposes, or because even 'indirect' training would require some form of gradient normalization, however, I would expect some clarity there.
> >
> > That is a great question. Indeed, intuitively, Batch Norm should not be needed since the CNN weights are generated. However, practically, we observed a slight degradation in results when the batch norm is removed.
> >
> > > On a similar matter, if the classifier is determined dynamically as in equations 4-6 with prototytpes, is there potentially a problem of scale, when adding many new classes, thus creating gradients of much larger magnitudes?
> >
> > That is a good point. Practically, we have not observed this happening even when the number of classes increases. In Fig.10 we show CHT learning with 20 tasks with 20 classes each (400 classes all together) without issues.
> >
> > > What exactly is the point of section 5? Does it give any new insight from MAML literature or is it more of a 'hey look what i found out' highlight?
> >
> > Section 5 provides connection of Prototypical Loss with MAML, which is one of the most known algorithms in few-shot learning. It serves as a potential explanation and justification of using Prototypical Loss in Hypertransformer.
> >
> > > How important is the order with which the classes and tasks are presented. Especially when not having a recurrent setting with shared weights, wouldn't this affect results significantly? I would expect at least an ablation study here.
> >
> > As we mention in the beginning of section 6, we train the model in episodic fashion where classes and samples for each task are sampled uniformly without replacement. So tasks and classes are considered iid from a given set of meta-training classes (except for section 6.3 where classes from each task are sampled from separate domains).

---

### Review · Reviewer_sN8z · 2024-01-04

**Summary Of Contributions:**

This paper presents an approach to continual few-shot learning based on a recurrent hyper-network. The hyper-network is trained to predict the weights of a CNN, given the previously set of predicted task weights and the support set of the new task. At inference time, such an approach would allow for gradient-free continual learning.

**Audience:**

Yes

**Claims And Evidence:**

No

**Requested Changes:**

Please address the issues I have discussed above

**Strengths And Weaknesses:**

I have broad issues with this paper that I hope the authors can either address via clarification or make changes to the paper to address

**Past work not properly contextualized through baselines / discussion**

[1] arguable does exactly what your paper claims to be doing but very little comparison to it is made in the paper.
The paper does mention other continual learning approaches like memory buffers for re-training and weight regularization methods but none of these are compared against. For example, [1], which is a very similar paper, compares to online EWC (elastic weight consolidation), which is a reasonable baseline to compare against.

Also, another point to note is that it is unclear that the the method the paper proposes actually does better than the simple baseline they implemented (ConstPN).  Figure 3 has no error bars around the plots. However, the deltas in performance are on the order of  <= 2%. Thus,  error bars would be great in order to gauge whether their results are actually significant.

**Limited discussion of weaknesses**

The method has several weaknesses that are not discussed including :
- complexity approach given the need to train a hyper-network
- the introduction of extra hyper-parameters due to the hyper-network
- memory and compute cost of training a hyper-network

**Implications of section 6.1**

I think I took away a different message from the results of section 6.1 than the authors present as reason the nature of their results.
First, note that all 3 types of models (a), (b), (c), produced near identical results. This to me indicates that the model is unable to perform positive forward transfer. Next, the authors say training on only S(1) or S(1) + S(2) produces poor results. This should not be surprising though since this is strictly less data than (a), (b) and (c) approaches have seen.  So I don't think this result is actually a testament to their method.

**Unsubstantiated claim**

In the last but 3rd paragraph of the introduction, you state that the models you learn with CHT *do not suffer from catastrophic forgetting*. This is a bold claim to make given that a) you only evaluate on task horizons up to 5  b)  Even for the Ominglot experiments in Fig 3, one can see if that if you generate $\theta_t$ beyond the number of training timesteps, $\theta_t$ performs worse than $\theta_i$ on task $i$, meaning that forgetting does occur when you extend the model past the training horizon


[1] CONTINUAL LEARNING WITH HYPERNETWORKS -- https://arxiv.org/pdf/1906.00695.pdf

---

> ### Author Response · Authors · 2024-01-19
> **Reply to reviewer sN8z**
>
> We thank the reviewer for taking the time to review our submission. We hope that the responses below will address and clarify the issues raised in your review.
>
> > Comparison against “Continual learning with hypernetworks” and online EWC
>
> Online EWC and “Continual learning with hypernetworks” and both great and competitive algorithms for continual learning. However, they are not specifically designed for few-shot learning. With so few examples available during the training time, methods such as EWC are not going to work well. Notice that even the very online EWC paper (Schwarz et al, 2018) says in their experimental setup (Section C.1) that they “are not treating Omniglot in the usual few-shot learning fashion”, precisely because their algorithm is not designed for few-shot learning.
>
> We have tried to use EWC for our settings and found it to work pretty bad even on simple problems, such as Omniglot. Please see the table in the general response above or see the updated results in the paper.
>
> Apart from not being a few-shot learning specific algorithm "Continual learning with hypernetworks" also has another problem. It learns task embeddings for specific tasks seen during training, limiting its ability to adapt to unseen tasks. In contrast, our method is trained on a few-shot learning task set and can generalize to novel few-shot tasks after training.
>
> > Adding error bars
>
> We agree that it is important to add confidence intervals to the experiments.  For both task- and class-incremental experiments (Figures 3 and 4), the confidence intervals do not exceed 0.5%. We have updated the paper with this information.
>
> > Limited discussion of weaknesses
>
> > complexity approach given the need to train a hyper-network;
> > memory and compute cost of training a hyper-network
>
> Indeed, the computational cost of training the hypernetwork can be quite large, as we train the network for 4 million steps to achieve the best results, which took us 12-20 hours using a single GPU.
>
> However, the training process is a one-time investment on a meta-training distribution of tasks. Once trained, the CHN can generate a target CNN network for new sets of tasks in a matter of seconds. For example, it took us 10 seconds to generate a CNN network for a novel 20 tasks of 20-way 1-shot Omniglot examples from Appendix A.4. This efficiency in generating new architectures is a key advantage when operating in dynamic task environments.
>
> > the introduction of extra hyper-parameters due to the hyper-network
>
> Indeed, HyperTransformer requires a separate Transformer network in order to generate a target CNN, whose architecture details and optimization parameters can be considered hyper-parameters of our method. In our work, we were able to reuse hyperparameters from the original HyperTransformer due to the similarity in evaluation datasets. However, in scenarios with novel datasets, practitioners may need to explore hyperparameter tuning to optimize performance.
>
> > Implications of section 6.1 and positive forward transfer
>
> What we have tried to show in section 6.1 is that each of the models (a), (b), and (c) has surprisingly similar performance to each other, even though they see data all at once (model (a)) or sequentially using mini-batches (models (b) and (c)).
>
> What it shows is that the models (b) and (c) are able to remember the information from the earlier iterations as if the full information was presented to it at once. We placed this experiment first to serve as a sanity check that our proposed model is actually using the information from the previous batches when learning on the new batch.
>
> Positive forward transfer generally refers to cases in which the performance on tasks learned earlier in the training improves after the model learns later tasks. In this particular experiment, the classes don't change from task to task (we train on mini-batches from the same 5-way classes), so we can't specifically check how much, for example, task 0 has improved after learning task 1.
>
> We demonstrated positive forward transfer in the later experiments. For example, on task-incremental learning on Omniglot, the performance of Task 0 improved from 86.31% to 86.55% after Task 1 was learned (Fig. 3, top-left plot, comparing orange and green markers on Task 0).
>
> > Unsubstantiated claim about catastrophic forgetting
>
> This is a very good point raised by the reviewer. Indeed, we have demonstrated that the CHT doesn’t suffer from catastrophic forgetting only within the settings that we have tried (up to 5 tasks). It remains to be seen if the catastrophic forgetting happens over a longer sequence of tasks. We have modified the manuscript to reflect this change.

---

### Author Response · Authors · 2024-01-19
**Comparison to baselines**

We thank the reviewers for their time and feedback. As a general comment for all the reviewers, we would like to discuss the comparison of our method with other baselines.

Our paper’s setting is different from most of the competing methods. Pure Continual Learning (CL) approaches won't work because they require substantial amounts of training data. Similarly, pure Few-Shot Learning (FSL) methods won’t work either, because the learning of new tasks will be impacted by catastrophic forgetting of previously learned tasks.

The closest setting to ours is Incremental Few-Shot Learning (IFSL), which first trains a base classifier using a large corpus (not using FSL) and then modifies that classifier to fit new few-shot tasks without causing forgetting. These methods are not applicable when the base classifier is absent. Our CHT method absorbs information from the training distribution into its weights $\psi$, enabling it to start from scratch and continually learn from the stream of FSL tasks. Another difference between our approach and IFSL is that IFSL methods need to be re-trained for a given stream of FSL tasks, while CHT generates the weights on the fly with no training required.

However, at the reviewer request, we have run the following baselines on 20-way, 1-shot Omniglot dataset using a network with 4 convolutional layers with 8 channels each (same setup as in Section 6.2 of the main paper):

1. _Pretraining_. Pretraining a network on a full training set, then fine-tuning it on an input sequence of tasks from the test set.
2. _MAML++_. As an example of a FSL method, we used MAML++ (Antoniou et al., 2019).
3. _EWC_. As an example of a CL method, we run Elastic Weight Consolidation (Kirkpatrick et al. 2017) on a pretrained network from above.

We evaluated these baselines on class-incremental settings, where the task is to predict both class and task attributes $p(\hat y=k, \tau|\hat x)$. As a reminder, task name indicates the range of tasks we used for evaluation (e.g. task name $0-3$ corresponds to four tasks from 0 to 3)

| Method |  |Task | name | | |
| -----------| :-----------: | :-----------: | :-----------: | :-----------: | :-----------: |
| | _0_    | _0-1_ | _0-2_ | _0-3_ | _0-4_ |
| _Pretraining_ |	38.4 |	21.1 |	8.3 |	4.8 |	3.9 |
| _MAML++_ |	81.4	| 58.2 |	33.5 |	24.4 |	19.8 |
| _EWC_ |	33.4	| 20.4	| 9.0 |	4.7 |	3.7 |
| _Continual HT (ours)_ |	87.2	| 80.0 |	75.1 |	76.8 |	69.3	|

None of the methods except for ours are able to achieve both CL and FSL performance at the same time.

---

### Decision · Action_Editor_8ZmQ · 2024-04-04

**Recommendation:** Accept with minor revision

**Comment:**

The proposed CHT method is a straightforward extension of the previously published HyperTransformer approach and therefore has relatively low novelty. It is important to note, however, that novelty is not a criterion for acceptance to TMLR. Comparison to baselines was also raised as an issue by reviewers. Due to the relative novelty of the continual few-shot learning setting, there are not clear established baselines in the literature to compare against. During the response period, experiments comparing to both few-shot and continual learning approaches were added and shown to perform much worse than the proposed CHT model.

Overall, the claims of this paper are somewhat modest, but are supported sufficiently by experiments and should be of interest to the community. Therefore, I recommend acceptance with the following minor revisions:
- Sufficient details regarding the HyperTransformer (as background material) and the Continual HyperTransformer should be added to allow for others to reproduce the results.
- Error bars should be added to the plots, especially Figures 3 and 4.

Additionally, the authors are encouraged to include the following modifications, which would improve the quality of the paper but are not critical to securing acceptance:
- Ablations showing the effects of (a) batch normalization and (b) the order of classes and tasks.
- Broader comparison to baseline continual learning methods, such as Online EWC (Schwarz et al. 2018) and “Continual Learning with Hypernetworks” (von Oswald et al. 2020). Although such approaches may not be specifically designed for the few-shot continual learning setting, it is still valuable to benchmark against them to demonstrate the challenges of the few-shot continual setting and the degree to which CHT is able to address the gap.

**Audience:**

There are two main arguments for the TMLR community to be interested in this work. The first is the interesting continual few-shot learning setting, which is a realistic scenario (as the paper points out) but has thus far seen relatively little work. The experiments could serve as a point of comparison for future work along these lines. The second is demonstrating that a transformer-based recurrent hypernetwork is effective in the continual few-shot setting.

**Claims And Evidence:**

This paper examines the continual few-shot learning setting in which a sequence of few-shot classification tasks are to be solved by a neural network. The paper introduces the Continual HyperTransformer (CHT), a recurrent transformer-based hypernetwork that directly predicts a set of weights for the current task, given the labeled support set and the weights from the previous task. The proposed CHT model extends the previously published HyperTransformer model to the continual few-shot setting. The experiments focus on two main settings: a task-incremental setting where task-level information is provided, and a class-incremental setting without task information. Experiments are presented on both Omniglot and TieredImageNet where the base classifier is a simple CNN.

There are several primary claims, each of which is supported by evidence:
1. The CHT model trained for class-incremental learning can also perform well in task-incremental learning. (Figure 6)
2. Catastrophic forgetting is not observed in CHT when evaluating on up to 5 tasks on the Omniglot and TieredImageNet datasets. (Figure 3)
3. The CHT model can be stopped at an intermediate timestep and effectively be applied to prior tasks. (Figures 3 & 4).
4. The CHT model can be applied to longer task sequences than seen during training. (Figures 9 & 10)